# A Study on the Injury Rate of Spanish Competitive Athletes as a Consequence of the COVID-19 Pandemic Lockdown

**DOI:** 10.3390/ijerph20010420

**Published:** 2022-12-27

**Authors:** José Manuel Prieto-Fresco, Daniel Medina-Rebollo, Jesús Fernández-Gavira, Antonio Muñoz-Llerena

**Affiliations:** 1Research Group “Social Inclusion, Physical Education and Sport, and European Policies in Research”, University of Seville, 41013 Seville, Spain; 2Physical Education and Sport Department, University School CEU San Pablo, 41013 Seville, Spain; 3Physical Education and Sport Department, University of Seville, 41013 Seville, Spain

**Keywords:** competitive sport, physical health, fitness, injury prevention

## Abstract

The COVID-19 lockdown may have had collateral effects on the conditions of athletes, with possible increases in injury risks upon the return to sports. Due to the existence of multiple factors of injury risks based on the lockdown and the return to sports, the aim of this study was to analyze the injury rates after the lockdown in competitive athletes. A non-experimental quantitative design based on a survey was carried out, and 94 athletes (42 males and 52 females; 23.57 ± 6.20 years) participated. Statistical analyses were carried out using non-parametric tests. Injury rates did not increase significantly after the lockdown, and there were no statistical differences between performance levels, types of sports, the design of a specific training program by sports professionals during the lockdown, the moment of injury, and the mechanism of injury. It is important to develop injury prevention protocols to prepare athletes after such a long period of detraining.

## 1. Introduction

The world has recently been hit by a pandemic originating from COVID-19, an infectious disease caused by severe acute respiratory syndrome coronavirus-2 (SARS-CoV-2), which was first detected in December 2019 in Wuhan, China [1]. Different restrictive measures were taken due to this pandemic, such as movement restrictions, social distancing, and the closures of sports facilities [2,3,4,5], which led to psychosocial sequels (e.g., anxiety, depression or insomnia) [6] and a decrease in sports participation and physical fitness [2,7,8]. These measures also compromised athlete training by limiting their physical activity, hindering their access to many forms of physical exercise and multidisciplinary sports teams (e.g., coaches, sports scientists, sports medicine professionals), and hampering team practices [4,9,10]. The reason for the closure was the concern about the elevated risk of COVID-19 infections in athletes due to the nature of competitive sports, i.e., close contact, tight locker rooms and training facilities, shared equipment, lack of social distancing, and the absence of face coverings during training and competitions [11]. A consequence of these restrictive measures was the interruption of sports competitions, disrupting sports and training routines [2,10,12,13,14,15]. In Spain, there was a lockdown enforced by health authorities, which lasted eight weeks and entailed a quarantine for the population, forbidding athletes from participating in physical activities outside of their homes [12,15].

The suspension of competitions and the ban on organized practices had significant impacts on the psychological and physiological conditions of athletes, leading to decreased performances [16]. On the one hand, some studies have shown that there was a relationship between isolation and an increase in different psychological and emotional disorders, such as depression or anxiety [6,17]. On the other hand, it was difficult for sports professionals to ensure that the training loads imposed on athletes by home training were adequate to reduce the potential losses of their training-induced physiological and morphological adaptations [18].

According to different authors, the restrictions in accessing sports facilities and the impossibility of participating in organized team training and competition would induce a total or partial loss of training-induced neuromuscular adaptations, a process known as detraining [19], resulting in a negative impact on the athlete’s capacity to perform sports-specific actions [3,12,16,20,21]. Detraining can have a negative effect on the neuromuscular system (i.e., muscle mass loss, damage to the neuromuscular junction), muscle protein metabolism (i.e., suppression of muscle protein synthesis), and cardiorespiratory system (i.e., reduction of VO_2max_, impairment of the steps along the O_2_ pathway) [22]. Examples of detraining effects can be found elsewhere [23,24].

Based on all of the previous information, it is evident that the COVID-19 pandemic lockdown could lead to an increased number of injuries upon the return to sports [16] as a result of the detraining process-related effect. In order to explore this issue, it is necessary to understand what an injury is and its possible classifications. This term could be defined as “any accident or physical dysfunction occurring during sport, or as a direct consequence of it” [25] (p. 42), which leads to the unavailability of the athlete to participate in future training or competitions [26]. According to [27], most injuries take place in the competition, with an injury rate (IR) of 23.8 injuries/1000 h, while in training, the IR was 3.4 injuries/1000 h. However, there are many discrepancies in the specifications of the IR throughout the literature, which makes it difficult to compare existing databases and studies [28]. To solve this problem, the latter authors define the concept of injury as “a term that applies to all processes that result in the destruction or alteration of the integrity of a tissue or part of the organism” (p. 8). Table 1 shows a classification that alludes to the type of injury and tissues affected, based on the orchard sports injury classification system (OSICS) [29,30,31].

The COVID-19 crisis has created many challenges, including an optimal and safe return to sports competition. In terms of risk factors for injuries during the lockdown (DL), the inevitable change of routine, the scarce sports equipment available (in most cases), and home training programs [32] should be highlighted, since they entailed a marked reduction in training specificity, intensity, frequency, and duration [9,11,33]. The latter can be considered the most relevant, as the athletes, accustomed to face-to-face training supervised by multidisciplinary teams of physical exercise professionals (which varies depending on the competitive level), began to train on their own, based on instructions provided by e-mail, video, or audio; they were required to adapt to these training sessions according to the reality of the space and the material available in each one [34].

To reduce the detraining effects and to carry out an optimal and safe return to sports training and competitions, a progressive and well-structured exercise program following the lockdown is essential [13]. Athletes usually trained at home DL, focusing on bodyweight-based and weightlifting strength training, as well as cardiovascular endurance conditioning [9,11,35]. However, home-exercise programs (characterized by limited equipment and facilities) do not provide enough stimuli to replicate the conditions of real competitions and usually lead to irregular training loads and a reduction in strength and general fitness levels [2,7,9,11,12,19,36,37].

Moreover, assuming that the athletes’ workloads were approximately 20 to 40% of their normal workloads, the recommended time to return to top-level training without a high risk of injury would be estimated at 3–5 weeks [38], although the return was further complicated by the extreme need to return to normalcy in an extremely short time. This was the case for many elite athletes, who experienced a reduction in rest time between competitions [2,10,11,13], as well as changes in nutrition, sleep, and exercise habits [7]. This situation resulted in an increase in IR, especially musculoskeletal injuries, upon return to sports at all competitive levels, mostly in the first two weeks after the return [2,3,10,11,13,16,39,40,41,42,43]. Due to the evidence of existing research about the multiple injury risk factors at the time of return to sport, which originate from the detraining effects caused by insufficient sports-specific training stimuli and the lack of organized training and competition DL, together with the inadequate return to training and competition, a research question arose:

To what extent has the IR changed after the lockdown (AL) when compared to before the lockdown (BL) in competing athletes?

Based on this question, the main objective of this research was to check whether there was increased IR AL in competitive athletes; we also analyzed the causality of the data in relation to the sports domain. The following research hypotheses arose from this objective:

**Hypothesis** **1** **(H1).**
*There will be a higher IR AL compared to the period BL.*


**Hypothesis** **2** **(H2).**
*The difference in the number of injuries between elite and amateur athletes increased AL.*


**Hypothesis** **3** **(H3).**
*People who practice contact sports will have greater tendencies to be injured compared to people who practice non-contact sports.*


**Hypothesis** **4** **(H4).**
*People whose training sessions were not designed by sports professionals DL had higher IRs AL than those whose training sessions were designed by sports professionals.*


**Hypothesis** **5** **(H5).**
*The difference between the number of injuries occurring in training and those occurring during competition increased in the AL period.*


**Hypothesis** **6** **(H6).**
*The type of exercise performed at home DL will influence the IR in the AL period.*


## 2. Materials and Methods

### 2.1. Study Design

In this study, we used a non-experimental quantitative design based on an online survey [44].

### 2.2. Sample

The target population of this research was athletes (men and women) over 18 years who practiced and competed in any sport (active practitioners, whether amateur or high-level athletes) in Spain. The selection criteria were that the athlete had to be (1) over the age of 18; and (2) an active practitioner in a competitive sport since eight months BL. A multi-phase sampling was carried out, there was no exact list of individuals who made up the target population, and there was no way of reaching these individuals directly. Therefore, members of the target population were linked to a grouping that could be sampled (in this case, by geographical areas and close contacts). Access to the sampling frame was achieved through convenience sampling, i.e., the sample consisted of a questionnaire in the Spanish language created in Google Forms.

The final sample was formed by 94 people, with a mean age of 23.57 ± 6.20 years, of whom, 42 were men (45%) and 52 were women (55%). The most practiced sport by the respondents was volleyball (*n* = 25, 27%), followed by soccer (*n* = 16, 17%), basketball (*n* = 13, 14%), and handball (*n* = 12, 13%). Other practiced sports were indoor soccer (*n* = 10, 11%), paddle tennis (*n* = 7, 7%), athletics (*n* = 6, 6%), kayaking (*n* = 2, 2%), swimming (*n* = 1, 1%), triathlon (*n* = 1, 1%), and tennis (*n* = 1, 1%). If we differentiate between both sexes, the most practiced sport by men was soccer (*n* = 15, 35.7%), while for women, it was volleyball (*n* = 18, 34.6%).

Furthermore, in terms of performance level, 41 persons (44%) considered themselves amateurs (i.e., competing at local and regional levels and for personal satisfaction), while 53 (56%) defined themselves as high-level athletes (i.e., competing professionally at a national or international level). Of the total sample, 88.3% (*n* = 83) claimed to have been injured at least once during their sports careers, of which, 53% (*n* = 44) were amateur athletes and 47% (*n* = 39) were high-level athletes.

### 2.3. Variables

Apart from the performance level (i.e., amateur and high-level athletes), which was already explained in the previous paragraph, there were different variables analyzed:The type of sport. It refers to the distinction of sports based on contact with the opponent. This variable has two levels: *contact sports* (i.e., sports that involve direct body contact with the opponent when performing, e.g., soccer, rugby, handball) vs. *non-contact sports* (i.e., sports that involve no direct body contact with the opponent; e.g., volleyball, tennis, swimming).Training design DL. It alludes to the professional design of the training sessions performed by athletes DL. This variable has two levels: *designed training* (i.e., training sessions that are individually designed and monitored by sports professionals, e.g., videoconferences during home training) vs. *non-designed training* (i.e., training sessions that are not designed by sports professionals and are performed autonomously by the athletes, e.g., YouTube fitness videos).Moment of injury. This variable describes the moment within the sports season when the athlete is injured. This variable has three levels: *training* (i.e., the athlete is only injured in a training session) vs. *competition* (the athlete is only injured in competition) vs. *training + competition* (the athlete is injured in both).Type of training DL. The present variable explains the types of exercises performed by the athletes DL. It includes four levels: *no exercise* (i.e., the athlete does not perform any exercise) vs. *endurance* (i.e., the athlete only performs endurance exercises, e.g., running, cycling) vs. *strength* (i.e., the athlete only performs strength and/or resistance exercises, e.g., weightlifting, eccentric training), vs. *strength + endurance* (i.e., the athlete combines both strength and endurance exercises).Mechanism of injury. The last analyzed variable is related to how the athlete is injured. It includes three levels: *overload* (i.e., there is no identifiable single external transfer of energy that causes the injury, but multiple accumulative bouts of energy transfer, e.g., tendinopathies, tennis elbow) vs. *trauma* (i.e., the injury is caused by an identifiable single external transfer of energy, e.g., concussions, dislocations) vs. *overload + trauma* (i.e., the athlete is injured at least once because of each mechanism).

### 2.4. Data Collection

The survey was created based on the injury register sheet developed in the FIFA Medical Assessment and Research Center (F-MARC) project [30]. The questionnaire is divided into four sections: the first section includes sociodemographic questions (i.e., age; gender; sport played; sport level; place of residence/competition; existence and number of previous injuries); the second section includes questions about injuries suffered in the eight months BL, from July 2019 to February 2020 (number and types of injuries); the third section is similar to the second section, but refers to the eight months AL; and the fourth section contains a series of questions about habits DL.

The survey was uploaded and shared on 2 March 2021 on the online platform Google Forms, by sending the link to complete it through different channels: email invitations; mailings to coaches and managers of local, regional, and national sports clubs; and dissemination on platforms, such as ResearchGate™, LinkedIn™, Facebook™, WhatsApp™, and Twitter™. After 15 days (17 March), the survey was shared again to collect more answers. Thirty days after the first sharing (1 April), the survey was closed; the results were registered in a Microsoft Excel sheet and included in the SPSS database.

The present study was conducted in accordance with the Declaration of Helsinki (1964) and was approved by the Research Ethics Committee of the University of Seville (internal code 1667-N-22). The survey included an introductory page describing the background and the aims of the survey, along with ethical information for participants. Survey participants were assured that all data would be used only for research purposes. Their answers are anonymous and confidential, following Google’s privacy policy (https://policies.google.com/privacy?hl=en (accessed on 19 December 2022). Participants were not allowed to provide their personal details at any point during the survey; furthermore, participants could withdraw from the survey at any time prior to submitting their responses, which were collected only by clicking on the “Submit” button. By completing the survey, participants gave their voluntary consent to participate in this study and were required to be honest with their responses [4].

### 2.5. Data Analysis

Following [45], after searching for missing and outlier values, descriptive statistics were calculated and a normality test (i.e., Kolmogorov–Smirnov) was carried out, showing a non-normal distribution. For checking differences between conditions, different tests were utilized: Mann–Whitney and Wilcoxon rank-sum tests for variables with only two conditions (i.e., high-level/amateur athletes, contact/non-contact sport, and designed/non-designed training program DL), and Kruskal–Wallis test for variables with three or more conditions (i.e., age, type of training DL, the moment of injury, and mechanism of injury). When significant differences were found between conditions in the Kruskal–Wallis test, an adjusted follow-up pairwise comparison was performed. For evaluating differences between the IR BL and AL, the Wilcoxon signed-rank test was used. Effect sizes were considered as small (*r* = 0.10), medium (*r* = 0.30), or large (*r* = 0.50), and were calculated based on the following equation:$$r=\frac{z-score}{\surd N}$$

Statistical analyses were carried out using SPSS software (26.0 version), and the level of significance was set at *p* = 0.05.

## 3. Results

The descriptive statistics for each level analyzed are shown in Table 2, and the total number of injuries and IRs for the levels BL and AL of the athletes are shown in Figure 1.

When differences between the two levels were compared (Table 3), the results showed no significant differences in the IR BL and AL between contact and non-contact sports. There were no significant differences between professionally-designed and non-designed training DL. Furthermore, high-level athletes suffered significantly more injuries than amateur athletes BL, but there were no significant differences between them AL. All effect sizes were small (*r* < 0.30).

The comparison of differences between three or more levels (Table 4) showed that the IR was significantly affected by the moment of injury (i.e., if the player was injured during training, competition, or both) and the mechanism of injury (i.e., if the player suffered an overload or a traumatic injury). However, the results showed that the age and the type of training performed DL had no effect on the IR.

Pairwise comparisons with adjusted *p*-values (Table 5) were used to analyze the differences between the levels included in the variables *Moment of injury* and *Mechanism of injury*. The analysis of the moment of injury showed that there were significant differences and large effect sizes (*r >* 0.50) between IRs when athletes were not injured compared to those who were injured during training, competition, or both. However, there were no differences between the remaining levels (i.e., injured in training, in competition, and both). Comparisons related to the mechanism of injury showed significant differences and large effect sizes (*r >* 0.50) between IRs when athletes did not suffer any injuries compared to those suffering from overload injuries, traumatic injuries, or both. However, there were no differences between the other three levels (i.e., suffering an overload injury, a traumatic injury, or both).

The differences between BL and AL IRs (Table 6) showed that the total number of injuries was not significantly different. Most of the variables (i.e., sex, performance level, type of sport, design of training DL, the moment of injury, and mechanism of injury) did not change significantly. Within the type of exercise performed DL, only the athletes who trained by combining strength and endurance exercises showed a significant reduction in their IR in the AL period when compared to BL; the IR of athletes who did not train or trained only with strength or endurance exercises did not significantly change. Effect sizes were small (*r* < 0.30) for most of the levels; medium (*r* > 0.30 and < 0.50) for the IRs in both training and competition and traumatic injuries; and large (*r* > 0.50) in both overload and traumatic injuries.

## 4. Discussion

The main objective of this research was to check whether there was a higher IR AL in competitive athletes when compared to the time BL, as well as to analyze the causality of the data in relation to the sports context. Regarding the first research question (to what extent has the IR changed AL when compared to BL in competing athletes?), after analyzing the data collected in the survey, it was observed that the IR did not increase significantly after the return to sports, but rather decreased among the respondents, although not significantly. The null hypothesis should be retained for H1–H5, since no significant differences were found between the different variables analyzed. Finally, H6 was fulfilled due to the significant reduction in the IR AL in the athletes who, DL, combined strength and endurance training.

The results obtained for the IRs, in general, are similar to those found in recent studies in competitive athletes [12,46], although there are studies that presented significant changes AL in college, amateur, and high-level athletes [10,16,39,41,42,43,47]. This difference may exist because a large part of the population was forced to stop competing in sports after the end of the quarantine due to various circumstances arising from the pandemic, such as confinements of 10 or 20 days as a consequence of direct contact with a COVID-19-positive person, positive testing, or the restrictive measures. In addition, many clubs (generally amateurs) chose not to form teams until the situation became clearer, reducing the number of competitions and, therefore, the requirements necessary to face them, which may have led to this small decrease in the number of injuries.

If we look further into this aspect, we could see that, based on the results of the survey, most athletes injured their ankles, knees, and thigh areas, with the most frequent injuries being sprains, muscle injuries, and tendinopathies, which is consistent with the existing literature [12,25,48]. The only injuries that increased AL were muscle injuries (by 14.5%) and tendinopathies (by 8.1%), indicating that there was an increase in the types of injuries that may be easier to prevent through individualized training. This may be caused by the fact that competent authorities did not allow different clubs to have long enough specific reconditioning periods for the athletes to recover from their neuromuscular and cardiorespiratory qualities, similar to what usually happens during pre-season after a transition period [3]. This is consistent with the results obtained in the study carried out by [39], who found that muscle injuries have been the most common among Bundesliga athletes after the resumption of the season; however, in the competitions that allowed for a longer reconditioning period (e.g., for the Spanish national soccer league, “La Liga”, when compared to Bundesliga), there were no significant differences between the IR BL and AL [12]. Therefore, coaches and trainers should be aware of this increased risk of injury (especially musculoskeletal injuries) and increase the subjective and objective monitoring of athletes.

When comparing the performance levels, the results showed that, although there were no statistical differences between both groups AL, high-level athletes suffered significantly higher numbers of injuries BL than amateur athletes. This may arise from the greater demands in training and competitions faced by high-level athletes [49], caused by higher internal (i.e., psychophysiological responses of the athletes that take place during an exercise) and external loads (i.e., loads prescribed by the coach to elicit psychophysiological responses) compared to amateur athletes [16,50]. Although several studies have shown significant increases in the number of injuries in high-level athletes, mainly due to the abrupt increase in the training load after returning to competition and the greater concentration of competitions in a short period of time [3,10,39,43,51,52,53], other studies have shown that changes in IRs did not significantly increase AL in high-level athletes [12,46], thanks to individualized work during this time. This difference could be attributed to the different conceptions of high-level athletes, which could lead to recruiting participants with different characteristics and, thus, different results. Additionally, amateur athletes did not see an increase in the IR AL either, as most of them stopped competing at the end of this period. The cessation of competition and, in many cases, of training, led to lower internal and external loads on these athletes and, therefore, fewer injuries.

In relation to the type of sport (contact or non-contact), despite the fact that there was a lack of significant differences in this variable, contact sports had slightly higher injury averages at both time points than non-contact sports, which may be logical, as the risk of injury from trauma is higher in contact sports. However, to the knowledge of the authors, there are no previous studies that have addressed the differences in the IR between contact and non-contact sports.

It was also observed that high-level athletes trained, to a greater extent, using training programs designed by sports professionals, when compared to amateur athletes (who, for the most part, trained on their own). However, the lockdown made it difficult for strength and conditioning coaches or other sports professionals to correct techniques, training intensities, or the speed of exercise executions [12,16]; moreover, the challenge was even greater given the extreme need to return to competitive normality in an extremely short time, as was the case in many elite sports [10,13,43]. This hindrance in maintaining high training intensity and specificity may negatively affect the optimization of endurance and strength adaptations since these factors are crucial for sports performances [9,54,55,56,57].

Although not significant, AL there was an increase in training injuries (17.1% more than in the BL period) compared to competition injuries. This increase in training injuries may be due to two main factors: the various suspensions of competitions (especially amateur) due to COVID-19 restrictions and the need to prepare athletes to reach their peak performances (in many cases, under time constraints) after such a long period of detraining. However, the results disagree with others in the literature showing a significant increase in the IR AL in Italian professional football players (in both training and competition) [43] and a higher IR in competition when compared to training [27].

Going deeper into the habits of athletes DL (and the sequences of these), most athletes either performed combined strength and endurance training or only strength training, leading to similar results to those found in other studies [7]. The significant decrease in the average IR of athletes who combined strength and endurance training may be because they worked on several physical capacities instead of focusing on just one, which may help the athletes to maintain their physical and physiological states for as long as possible, bearing in mind that detraining is very difficult to avoid due to the restrictive environments, where most athletes have found themselves during social isolation, and the impossibility of receiving sports-specific training stimuli [9,12,37]. Some studies have suggested that training during home lockdowns should focus on low loads and high volume contractions to increase protein synthesis [22,58] and a combination of endurance training through HIIT (high-intensity interval training) and strength and power training, mainly through eccentric work, to reduce muscle damage when restarting competitions [22,59].

### 4.1. Theoretical and Practical Implications

This research has both theoretical and practical implications. Regarding the theory, the present study delves into the effects of a lockdown (or a prolonged irregular training period) on Spanish athletes and their return to sports.

From a practical perspective, this work gives an overview of what Spanish competitive athletes have experienced DL and AL, helping sports professionals, clubs, policymakers, and stakeholders to understand their situations and needs when a prolonged stoppage of regular training scheduling occurs. A few recommendations for all agents involved in sports competitions have emerged from the results of this research and the existing literature [9,11], to help athletes safely return to sports:Provide enough resources for athletes to carry out remote training properly.Prepare training facilities to allow permanent access.Implement standardized injury prevention practices (e.g., scientifically-sound training programs DL and return to sports).Create and organize a multidisciplinary network of social agents (including professionals, families, and peers) for supporting athletes.Promote psychological training and caring about mental health.

### 4.2. Limitations and Future Lines of Research

With regard to the limitations encountered, this study would have been even more reliable if it had been possible to reach a larger number of athletes and, in this way, expand the sample to observe whether there might be any changes in the results obtained, but due to the global pandemic in which society was immersed, the possibility of choosing another type of sampling was reduced; we carried out a convenience sampling based on the dissemination of a questionnaire via the internet. Another limitation is that most of the athletes participated in team sports (only 11 were involved in individual sports), which could have affected the results due to the differences in internal and external loads between individual and team sports.

Finally, regarding future lines of research, it would be interesting to carry out more studies on this subject in order to reach a consensus on the consequences that a longer than normal period of social isolation and sports stoppage (more than 3 months) can have on the lives of athletes and the injury risks during the return to sports, in order to develop standardized protocols for the prevention of injuries. Another insightful approach would be the complementation of quantitative studies with qualitative research, analyzing the experiences and perceptions of athletes DL (or longer irregular training periods) to provide more insight into the return to the sports field from the athlete’s perspective. Finally, it would also be of great importance for different institutions to offer longer periods of preparation for athletes to counteract detraining and for a less hurried return to competition, to allow athletes to return at optimal levels of performance.

## 5. Conclusions

IRs did not significantly increase AL, and there were no significant differences between sports levels, type of sport, the design of the training program by sports professionals DL, the time of injury, and the mechanism of injury. In contrast, IRs were significantly reduced in athletes who performed combined strength and cardiorespiratory endurance training DL. Therefore, it is essential to develop standardized injury prevention protocols to prepare athletes both during and after a longer-than-usual period of detraining, such as the COVID-19 lockdown in 2020.

## Figures and Tables

**Figure 1 ijerph-20-00420-f001:**
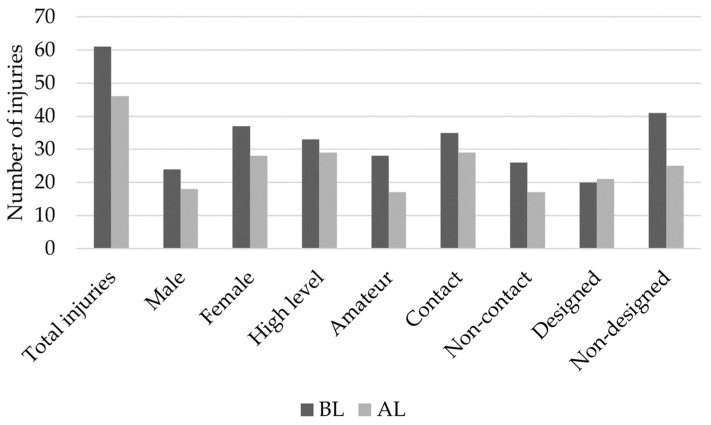
Number of injuries (BL and AL) for the different levels of examined variables.

**Table 1 ijerph-20-00420-t001:** Classification according to the type of injury and tissue affected.

Main Category	Subcategories
Fractures and bone stress	Bone fractures Other bone injuries
Joints and ligaments	Dislocation/subluxation Ligament injury Meniscus/articular cartilage injury Synovitis
Muscles and tendons	Muscle rupture/tear/spasm/cramp Tendon tear/tendinopathy/bursitis
Skin	Hematoma/contusion Abrasion/laceration
Central/peripheral nervous system	Concussion Structural brain injury Nerve injury Eye injury
Other	Visceral Dental injury Undiagnosed injury

*Note*. Adapted from [29,30,31].

**Table 2 ijerph-20-00420-t002:** Descriptive statistics of IRs BL and AL.

Injuries	N		M		SD		Skewness		Kurtosis	
	BL	AL	BL	AL	BL	AL	BL	AL	BL	AL
Total	94	94	0.65	0.49	±0.839	±0.864	1.303	2.534	1.756	8.388
Athlete (male)	42	42	0.57	0.43	±0.991	±0.887	1.764	3.635	2.652	17.030
Athlete (female)	52	52	0.71	0.54	±0.696	±0.851	0.461	1.663	−0.820	2.168
Athlete (high-level)	41	41	0.80	0.71	±0.782	±1.078	0.694	2.136	0.045	5.558
Athlete (amateur)	53	53	0.53	0.32	±0.868	±0.613	1.833	2.293	3.703	6.357
Athlete (contact sport)	51	51	0.69	0.57	±0.787	±0.944	0.885	2.620	0.055	9.245
Athlete (non-contact sport)	43	43	0.60	0.40	±0.903	±0.760	1.704	2.247	3.345	5.046
Training (designed)	29	29	0.69	0.72	±0.761	±1.131	0.606	2.342	−0.972	6.694
Training (non-designed)	65	65	0.63	0.38	±0.876	±0.700	1.526	2.117	2.521	4.729
Injury (training)	21	22	1.29	1.32	±0.463	±0.646	1.023	1.924	−1.064	2.631
Injury (competition)	15	7	1.20	1.00	±0.561	±0.000	2.919	-	8.388	-
Injury (training plus competition)	7	3	2.29	3.33	±0.756	±1.528	2.646	0.935	7.000	-
Injury (no exercise)	14	14	0.43	0.29	±0.646	±0.469	1.303	1.067	0.951	−1.034
Injury (endurance)	7	7	0.43	0.57	±0.787	±1.134	1.760	2.156	2.361	4.580
Injury (strength)	31	31	0.61	0.65	±0.761	±1.170	0.806	2.359	−0.756	5.965
Injury (strength plus endurance)	42	42	0.79	0.43	±0.951	±0.630	1.348	1.203	1.999	0.433
Injury (overload)	20	19	1.25	1.26	±0.444	±0.562	1.251	2.158	−0.497	4.253
Injury (trauma)	18	10	1.33	1.10	±0.594	±0.316	1.683	3.162	2.219	10.000
Injury (overload plus trauma)	5	3	2.40	3.67	±0.894	±1.155	2.236	1.732	5.000	-

*Note*. N, sample; M, mean; SD, standard deviation.

**Table 3 ijerph-20-00420-t003:** Comparison of IR differences between variables with two levels (BL and AL).

Conditions	*U*	*W_s_*	Mean		*z-Score*	*r*	*p*
			L1	L2			
Male/female	1302.50	2680.50	0.50	0.63	1.777	0.18	0.076
High-level/amateur (pre)	1343.00	2204.00	0.80	0.53	2.171	0.22	0.030 *
Contact/non-contact (pre)	1198.00	2524.00	0.69	0.60	0.855	0.09	0.392
Designed/non-designed (pre)	1014.00	1449.00	0.69	0.63	0.650	0.07	0.516
High-level/amateur (post)	1301.00	2162.00	0.71	0.32	1.960	0.20	0.050
Contact/non-contact (post)	1219.50	2545.50	0.57	0.40	1.119	0.12	0.263
Designed/non-designed (post)	1102.50	1537.50	0.72	0.38	1.570	0.16	0.117

*Note*. *U*, Mann–Whitney U; *W_s_*, Wilcoxon’s W; L, level; *r*, effect size; *p*, asymptotic significance; * *p* < 0.05.

**Table 4 ijerph-20-00420-t004:** Comparison of IRs between variables with three or more levels of BL and AL.

Levels	*H*	Degrees of Freedom	*p*
Age—pre	28.942	19	0.067
Age—post	14.079	19	0.779
Type of training DL—pre	2.266	3	0.519
Type of training DL—post	0.532	3	0.912
Moment of injury—pre	88.232	3	0.000 *
Moment of injury—post	91.174	3	0.000 *
Mechanism of injury—pre	87.371	3	0.000 *
Mechanism of injury—post	91.288	3	0.000 *

*Note*. *H*, Kruskal–Wallis test statistic; *p*, asymptotic significance; * *p* < 0.05.

**Table 5 ijerph-20-00420-t005:** Pairwise comparisons for the different levels of the variables ‘moment of injury’ and ‘mechanism of injury’ between BL and AL periods.

Levels	*p*		*r*	
	BL	AL	BL	AL
*Moment of injury*				
None—training	0.000 *	0.000 *	−0.84	−0.90
None—competition	0.000 *	0.000 *	−0.73	−0.57
None—both	0.000 *	0.000 *	−0.81	−0.55
Training—competition	1.000	1.000	0.05	0.07
Training—both	0.846	1.000	−0.28	−0.19
Competition—both	0.606	1.000	−0.35	−0.34
*Mechanism of injury*				
None—overload	0.000 *	0.000 *	−0.82	−0.86
None—traumatic	0.000 *	0.000 *	−0.81	−0.67
None—both	0.000 *	0.000 *	−0.71	−0.56
Overload—traumatic	1.000	1.000	−0.02	0.04
Overload—both	1.000	1.000	−0.28	−0.23
Traumatic—both	1.000	1.000	−0.27	−0.31

*Note*. *p*, adjusted significance; *r*, effect size; * *p* < 0.05.

**Table 6 ijerph-20-00420-t006:** Comparison of differences in the IRs for the examined variables between BL and AL periods.

**Injuries**	** *T* **	**Mean**		** *z-Score* **	** *r* **	** *p* **
		**BL**	**AL**			
Total	360.00	0.65	0.49	−1.663	−0.12	0.096
Athlete (male)	74.50	0.57	0.43	−0.858	−0.09	0.391
Athlete (female)	110.50	0.71	0.54	−1.493	−0.15	0.136
Athlete (high-level)	104.00	0.80	0.71	−0.773	−0.09	0.439
Athlete (amateur)	80.50	0.53	0.32	−1.566	−0.15	0.117
Athlete (contact)	147.00	0.69	0.57	−1.092	−0.11	0.275
Athlete (non-contact)	51.00	0.60	0.40	−1.240	−0.13	0.215
Training type (designed)	31.50	0.69	0.72	−0.136	−0.02	0.892
Training type (non-designed)	183.50	0.63	0.38	−1.859	−0.16	0.063
Injury (training)	18.00	1.29	1.32	0.000	0.00	1.000
Injury (competition)	0.00	1.20	1.00	−1.000	−0.21	0.317
Injury (training plus competition)	3.00	2.29	3.33	1.342	0.42	0.180
Injury (no exercise)	13.50	0.43	0.29	−0.707	−0.13	0.480
Injury (endurance)	1.00	0.43	0.57	1.00	0.27	0.317
Injury (strength)	51.00	0.61	0.65	−0.96	−0.12	0.923
Injury (strength plus endurance)	59.50	0.79	0.43	−2.055	−0.22	0.040 *
Injury (overload)	20.00	1.25	1.26	0.302	0.05	0.763
Injury (trauma)	0.00	1.33	1.10	−1.732	−0.33	0.083
Injury (overload plus trauma)	6.00	2.40	3.67	1.633	0.58	0.102

*Note*. T, test statistic; *r*, effect size; *p*, asymptotic significance; * *p* < 0.05.

## Data Availability

The data presented in this study are available upon request from the corresponding author.

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
