# Peer review of "A Study on the Injury Rate of Spanish Competitive Athletes as a Consequence of the COVID-19 Pandemic Lockdown"

_ijerph, 2022, doi:10.3390/ijerph20010420_

Round 1

Reviewer 1 Report

Title: Since no significant effects were found on the outcome parameters, it is suggested that the title is modified. May be: "A study of the injury rate of Spanish competing athletes as a consequence of COVID-19 pandemic lockdown".

Introduction

Lines 36-37: lack of space and inability to perform sport-specific exercises at home? A clarification is needed here.

Line 38: please correct to "upon return...".

Lines 86-87: H4: in my opinion not enough data have been provided for in the introduction to substantiate this hypothesis. I suggest that it is removed.

Lines 88-90: the purpose of the study is based on the assumption -and fact to a great extent- that due to non being supervised by their coaches and sport-related professionals, athletes would have a higher injury rate upon return to sport and competition. Therefore, this hypothesis seems contradictory to the aim of the study. Or do they authors refer to elit athletes, for whom the "distancing" rules were bent and they had supervised training? this needs to be clarified.

Lines 91-92: H6: what is the difference between this hypothesis and H1? It is recommended that it is removed.

Line 93-94: H7: what is the point of this hypothesis? The authors have not used different training methods that could lead to forming a hypothesis. Rather, due to the fact that training was not carried out in the usual conditions, the effect of detraining or not adequate training stimuli caused concern for higher injury rates after lockdown was ceased.

Materials and methods

Lines 98-103: it is suggested that these lines are removed or reduced. The first sentence has all the information.

Lines 160-161: same comment as for lines 88-90. 

Line 165: the term "related samples" is a bit confusing, please clarify.

Results

Table 1 (same comment for the other tables)

The format of Table 1 could be improved. It is suggested that "injuries" is written as the title for the 1st row and not repeated. Then, if the different fixed factors could be grouped in 2-3 groups, for example 1st group could be labelled "athlete" including gender, level and type of sport and 2nd group "training type" and may be 3rd group "injury" in order to make the Table easier to read.

Line 187: it should be reported in the statistical analysis section that adjusted or corrected (with what test?) post hoc comparisons with  were performed.

Lines 192-199: it is suggested that these results are reported in a more clear way. in their present condition, they are confusing. It is also recommended that the various levels of the fixed factors, for example mechanism of injury be described in detail in the statistics section. Also, perhaps the authors could give an example of tan injury from the category both overload and traumatic injury. if there was an underlying condition of overuse and then a traumatic event happened, it is difficult to determine the exact cause of injury. So, may be this category be omitted.

Lines 208-210: the conditions cannot change. the authors probably mean that the rate or number of injuries as classified per fixed factor were not different between pre and post lockdown. They need to re-write it correctly.

Discussion

Overall comment on discussion: in its present form, 1/3 of the discussion is a repetition of the results. In my opinion, rejection or not of the study's hypotheses should be reported in one paragraph at the start of the discussion in a way that summarizes all the non-significant differences between pre- and post-lockdown. Then, the next paragraphs could be used to discuss the reasons of why differences were not found.  

Lines 223-224: "In this section.... exposed": unnecessary, it could be deleted.

Line 229: correct to "...did not increase..."

Lines 233-234: "...that disagree ...elite athletes". this phrase needs to be written in more correct syntax.

Lines 249-250: it is reasonable that high level athletes would face a greater internal and external load however the authors need to provide references for this statement. Also, it is suggested that they define the internal and external load so that it becomes clear what do they mean.

Line 333, section 4.2.: about limitations: besides the restrained sample size due to reasons well-argued by the authors, another limitation is the fact that respondents practiced only team sports. This could have affected the results because although with the exception of volleyball, the other 3 sports were contact sports, the training load is not the same as compared to non-team sport. So, it is suggested that the authors address this point as well.

Author Response

Dear reviewer,

Thank you for your suggestions and considerations about this research, they have been very useful in improving the quality of the manuscript. Now, we will try to answer in the best way possible your comments, although we have struggled a little to answer some of the comments because we received five different revisions, and some of them were contradictory. We apologize in advance if we have not been able to fulfill all the reviews.  

The lines we use are those of the clean version of the manuscript.

  1. Title: Since no significant effects were found on the outcome parameters, it is suggested that the title is modified. Maybe: "A study of the injury rate of Spanish competing athletes as a consequence of COVID-19 pandemic lockdown".
    1. Title has been changed (L2-3)
  2. Lines 36-37: lack of space and inability to perform sport-specific exercises at home? A clarification is needed here.
    1. Introduction has been rewritten and extended, and this sentence has been modified for better understanding (L32-51)
  3. Line 38: please correct to "upon return...".
    1. Correction has been made (L62)
  4. Lines 86-87: H4: in my opinion not enough data have been provided for in the introduction to substantiate this hypothesis. I suggest that it is removed.
    1. H4 has been removed
  5. Lines 88-90: the purpose of the study is based on the assumption -and fact to a great extent- that due to non being supervised by their coaches and sport-related professionals, athletes would have a higher injury rate upon return to sport and competition. Therefore, this hypothesis seems contradictory to the aim of the study. Or do they authors refer to elite athletes, for whom the "distancing" rules were bent and they had supervised training? this needs to be clarified.
    1. Changes have been made to this hypothesis (H4, former H5) to clarify its meaning.
  6. Lines 91-92: H6: what is the difference between this hypothesis and H1? It is recommended that it is removed.
    1. This hypothesis refers to the possibility that there might be differences between injuries suffered in training and in competition, while H1 alludes to injury rate in general, without differencing the moment of injury. We believe this hypothesis may help to provide additional insight to the field of study.
  7. Line 93-94: H7: what is the point of this hypothesis? The authors have not used different training methods that could lead to forming a hypothesis. Rather, due to the fact that training was not carried out in the usual conditions, the effect of detraining or not adequate training stimuli caused concern for higher injury rates after lockdown was ceased.
    1. According to some of the references included in the document, both in the introduction and discussion (L87-94 and 354-367), the type of exercise performed during the lockdown could be relevant to determine the probability of reducing (or not) injury rates. In fact, results showed a significant reduction in the number of injuries of athletes who performed a combination of strength+endurance exercise during lockdown, when compared to no exercise, only endurance exercises or only strength exercises.
  8. Materials and methods. Lines 98-103: it is suggested that these lines are removed or reduced. The first sentence has all the information.
    1. Lines have been removed (L125-126)
  9. Lines 160-161: same comment as for lines 88-90.
    1. The whole section “Data analysis” has been modified (L213-227). The variables have been explained in a new section called “Variables” (L153-184).
  10. Line 165: the term "related samples" is a bit confusing, please clarify.
    1. The name of the test has been clarified (L222-223).
  11. Table 1 (same comment for the other tables). The format of Table 1 could be improved. It is suggested that "injuries" is written as the title for the 1st row and not repeated. Then, if the different fixed factors could be grouped in 2-3 groups, for example 1st group could be labelled "athlete" including gender, level and type of sport and 2nd group "training type" and may be 3rd group "injury" in order to make the Table easier to read.
    1. Tables 2 (L232) and 7 (now, table 6) (L275) have been modified.
  12. Line 187: it should be reported in the statistical analysis section that adjusted or corrected (with what test?) post hoc comparisons with were performed.
    1. Information about follow-up comparisons has been included (L221-222).
  13. Lines 192-199: it is suggested that these results are reported in a more clear way. in their present condition, they are confusing. It is also recommended that the various levels of the fixed factors, for example mechanism of injury be described in detail in the statistics section. Also, perhaps the authors could give an example of tan injury from the category both overload and traumatic injury. if there was an underlying condition of overuse and then a traumatic event happened, it is difficult to determine the exact cause of injury. So, may be this category be omitted.
    1. Results have been structured in a clearer way.
    2. The different levels of fixed factors have been described in the “Variables” section (L153-184).
    3. We believe it is interesting to differentiate between athletes who only got injured because of overload or trauma and those who got injured because of both (at least one injury from each type). This information could be helpful to analyze the injury rate from a broader perspective.
  14. Lines 208-210: the conditions cannot change. the authors probably mean that the rate or number of injuries as classified per fixed factor were not different between pre and post lockdown. They need to re-write it correctly.
    1. The sentence has been rewritten (L265-266)
  15. Overall comment on discussion: in its present form, 1/3 of the discussion is a repetition of the results. In my opinion, rejection or not of the study's hypotheses should be reported in one paragraph at the start of the discussion in a way that summarizes all the non-significant differences between pre- and post-lockdown. Then, the next paragraphs could be used to discuss the reasons of why differences were not found.
    1. A change in the discussion's structure has been made (L277-367).
  16. Lines 223-224: "In this section.... exposed": unnecessary, it could be deleted.
    1. The sentence has been deleted (L280)
  17. Line 229: correct to "...did not increase..."
    1. Correction has been made (L282)
  18. Lines 233-234: "...that disagree ...elite athletes". this phrase needs to be written in more correct syntax.
    1. The sentence has been rewritten (L289-290)
  19. Lines 249-250: it is reasonable that high level athletes would face a greater internal and external load however the authors need to provide references for this statement. Also, it is suggested that they define the internal and external load so that it becomes clear what do they mean.
    1. References and definitions about internal and external load have been included (L318-320)
  20. Line 333, section 4.2.: about limitations: besides the restrained sample size due to reasons well-argued by the authors, another limitation is the fact that respondents practiced only team sports. This could have affected the results because although with the exception of volleyball, the other 3 sports were contact sports, the training load is not the same as compared to non-team sport. So, it is suggested that the authors address this point as well.
    1. There were athletes who practiced individual sports, but we only reported the most practiced ones. We have included the whole list of sports practiced by the participants (L140-146), and we have added in the limitations section that the majority of the participants competed in team sports (L394-397).

We hope we have answered properly all your considerations. Again, thank you for your time and effort in reviewing our manuscript. Best regards.

Author Response

Dear reviewer,

Thank you for your suggestions and considerations about this research, they have been very useful in improving the quality of the manuscript. Now, we will try to answer in the best way possible your comments, although we have struggled a little to answer some of the comments because we received five different revisions, and some of them were contradictory. We apologize in advance if we have not been able to fulfill all the reviews. 

The lines we use are those of the clean version of the manuscript.

  1. Abstract: BMI data will be useful
    1. We do not really understand what you mean by BMI. We do not see any relation between Body Mass Index and this study, and we do not have enough information about the athletes to calculate it. If BMI refers to some other variable, please let us know and we will try to include it in further revisions.
  2. Introduction: Perhaps the authors could allude to some of the most affected physiological markers to further explain the detraining phenomenon.
    1. Physiological markers and other relevant information about detraining have been included (L52-60)
  3. Table 1: The referenced classification of injuries is very specific to a sport modality, being possibly not entirely adequate for such a generic classification as the one you want to show in the text as a whole.
    1. The text has been rewritten. This classification is the Orchard Sports Injury Classification System (OSICS). Although the original reference we used was from tennis, it is a general classification that has been utilized in several studies. New references have been added (L73-76).
  4. Materials and Methods. The study design is very “theoretical”
    1. The study design has been simplified (L125-126)
  5. What were the criteria for distinguishing the sample categories? (amateur or high level) (line 120)
    1. Definitions of high level and amateur athletes have been included in L148-150. Definitions of the remaining variables analyzed have been provided in L153-184.
  6. The survey protocol is adapted from a study designed far from the pandemic conditions. Do you think is really appropriated? (line 126)
    1. The survey protocol has been utilized in several studies on injury rates in different sports, and it was designed by experts in the field. We believe that the protocol is appropriate since we did not modify anything, athletes only had to fulfill it twice, once for the pre-lockdown period and once for the post-lockdown period.
  7. The data analysis paragraph is written in a style not suitable for this type of publication, more similar to that of a university paper than to that of a publication such as this journal. The authors are kindly requested to rewrite it.
    1. The data analysis section has been rewritten (L213-227)
  8. The numbering of the tables is incorrect, please revise them.
    1. The table numbers have been revised.
  9. Median data of table “Comparison of injury rate differences between two conditions in pre- and post-lockdown” has no sense.
    1. Median values have been changed to mean values. We followed the protocol of non-parametric tests by Field [45], who explains that the median is more accurate and correct when performing non-parametric tests than the mean. That is the reason why we used the median instead of the mean.
  10. The overall presentation of the results is very confusing for the reader.
    1. The presentation of the results has been restructured.
  11. It is suggested that the wording of the discussion be rewritten. Several concepts are repeated throughout the text.
    1. The discussion has been reviewed and repeated concepts have been changed.
  12. The absence of references to results that help to frame some of the statements in the discussion makes it difficult to understand this section.
    1. More references related to previous results and theoretical information have been included throughout the discussion.
  13. General comments. The use of abbreviations and acronyms could be improved throughout the text to facilitate reading.
    1. Abbreviations have been included for injury rate (IR), before lockdown (BL), after lockdown (AL), and during lockdown (DL)

We hope we have answered properly all your considerations. Again, thank you for your time and effort in reviewing our manuscript. Best regards. 

Reviewer 3 Report

Thanks to the authors for submitting this manuscript and the editors for the opportunity to review this work.

The title of the study was “Effects of COVID-19 pandemic lockdown on the injury rate of Spanish competing athletes”.

The study was to analyze the injury rates after lockdown in competitive athletes.

The strong part is that it is a relevant topic for basic research. The weak part is that it is the sample count which 94 participants using online form it seems a few for these interesting investigations.

Overall, you had a good and interesting manuscript.

-          Your introduction, material, and methods sounded good.

-          The results section was well organized and can be improved by using graphs.

Author Response

Dear reviewer,

Thank you for your suggestions and considerations about this research, they have been very useful in improving the quality of the manuscript. Now, we will try to answer in the best way possible your comments, although we have struggled a little to answer some of the comments because we received five different revisions, and some of them were contradictory. We apologize in advance if we have not been able to fulfill all the reviews. 

The lines we use are those of the clean version of the manuscript.

  1. The strong part is that it is a relevant topic for basic research. The weak part is that it is the sample count which 94 participants using online form it seems a few for these interesting investigations.
    1. We carried out a statistical power analysis using G-Power 3.1 software, and the result was that we needed a sample size of 92 participants. In any case, we will keep your suggestion in mind for future research, thank you very much.
  2. The results section was well organized and can be improved by using graphs.
    1. Figure 1 has been included (L234)

We hope we have answered properly all your considerations. Again, thank you for your time and effort in reviewing our manuscript. Best regards.

Reviewer 4 Report

Thank you for the possibility to review the manuscript „Effects of COVID-19 pandemic lockdown on the injury rate of Spanish competing athletes“. The manuscript presents a novel study analyzing the crucial factors of injury risk after the lockdown. However, several considerations should be address in the manuscript.

Line 120: add the definition of amateur and high level athlete. The difference in the high level athlete definitions can be the reason for the discrepancy of results between the manuscript and previous studies.

Line 123: add information about the country where the high level athletes perform.

Line 125: add the information about the language of the survey.

Line 132: add information about the restriction in Spain during the lockdown period.

Table 1: Consider showing the injury rates during pre-lockdown and post-lockdown for amateur and high level athletes separately

Results: Consider additional analysis of the time period between lockdown and the first post-lockdown competition and the injury rate.

Author Response

Dear reviewer,

Thank you for your suggestions and considerations about this research, they have been very useful in improving the quality of the manuscript. Now, we will try to answer in the best way possible your comments, although we have struggled a little to answer some of the comments because we received five different revisions, and some of them were contradictory. We apologize in advance if we have not been able to fulfill all the reviews. 

The lines we use are those of the clean version of the manuscript.

  1. Line 120: add the definition of amateur and high level athlete. The difference in the high level athlete definitions can be the reason for the discrepancy of results between the manuscript and previous studies.
    1. Definition of amateur and high level athlete has been included (L148-150)
  2. Line 123: add information about the country where the high level athletes perform.
    1. Information about the country (Spain) has been included (L130)
  3. Line 125: add the information about the language of the survey.
    1. Information about the language (Spanish) has been included (L138)
  4. Line 132: add information about the restriction in Spain during the lockdown period.
    1. Information about restrictions in Spain during confinement has been included (L41-43)
  5. Table 1: Consider showing the injury rates during pre-lockdown and post-lockdown for amateur and high level athletes separately
    1. Figure 1 has been included. It shows the total number of injuries of each variable analyzed (L234)
  6. Results: Consider additional analysis of the time period between lockdown and the first post-lockdown competition and the injury rate.
    1. We are not sure what kind of analysis you are requesting us. We have carried out all the analyses needed to check the different hypotheses of the study, and we are not sure what different analyses would provide more insight regarding the objective, research question, and hypotheses.

We hope we have answered properly all your considerations. Again, thank you for your time and effort in reviewing our manuscript. Best regards.

Reviewer 5 Report

Dear authors,

I begin by congratulating you by work and the relevance of the theme. Than you for the opportunity to read this paper The results they find are very interesting and deserve reflection.

However, the article lacks more theoretical investment. 

This is a hypothetical deductive study, which presents on page 3 seven hypotheses of investigation. However, these hypotheses do not present theoretical support, and I believe each hypothesis must be accompanied by its theoretical basis to understand the authors' arguments. 

Concerning the method, the study also lacks some key elements. Authors should inform about when and during when they have performed data collection and disclose the inclusion and exclusion criteria of participants. 

Finally, the lack of theory is reflected again in the discussion. The authors should relate their findings to the previous theory and discuss the theoretical and practical implications that are currently unclear. 

Strengthening the quality and pertinence of the study, however, its contribution to science is limited if no theoretical strengthening effort of the article is made. 

Thank you for your attention and opportunity. 

Author Response

Dear reviewer,

Thank you for your suggestions and considerations about this research, they have been very useful in improving the quality of the manuscript. Now, we will try to answer in the best way possible your comments, although we have struggled a little to answer some of the comments because we received five different revisions, and some of them were contradictory. We apologize in advance if we have not been able to fulfill all the reviews. 

The lines we use are those of the clean version of the manuscript.

  1. The article lacks more theoretical investment. This is a hypothetical deductive study, which presents on page 3 seven hypotheses of investigation. However, these hypotheses do not present theoretical support, and I believe each hypothesis must be accompanied by its theoretical basis to understand the authors' arguments.
    1. The introduction has been rewritten and expanded, including more references and theoretical background to support the hypotheses.
  2. Concerning the method, the study also lacks some key elements. Authors should inform about when and during when they have performed data collection and disclose the inclusion and exclusion criteria of participants.
    1. Data collection dates have been included in L194-200.
    2. Selection criteria have been better explained in L130-132.
  3. Finally, the lack of theory is reflected again in the discussion. The authors should relate their findings to the previous theory and discuss the theoretical and practical implications that are currently unclear.
    1. The discussion has also been rewritten and expanded, including more references about the relation of the present study with other studies and with the theoretical background of the field of study.
    2. A section called “Theoretical and practical implications” has been included (L368-387)

We hope we have answered properly all your considerations. Again, thank you for your time and effort in reviewing our manuscript. Best regards.

Round 2

Reviewer 1 Report

The authors have greatly improved the first version of the manuscript. 

Upon reviewing it for the 2nd time, I have some comments and corrections, mostly grammar-related.

Comments

page 2., line 52: correct "training load caused to..." to "training load imposed on ..."

page 2, line 67: it is suggested that is written "...return to sport, as a result of the detraining process related effect.

page 2, line 88: correct to "alludes"

page 4, line 138 & 140: better state "sport professionals"

page 5, section 2.3. This is helpful, however it is recommended that is written in a statistical way, whenever possible. For example:

Type of sport: variable with 2 levels, specifically contact (....) vs non-contact sports (.....)

page 7, line 276: add at the end of the phrase ... and level of significance was set at a=0.05.

general comment about results: statistically speaking, in my opinion, it would be more accurate if the variable referred to two or three levels instead of conditions. In the sense that a condition implies a situation that could be modified.

Figure 1, legend: it is suggested to change it to: Number of injuries before (BL) and after lockdown (AL) for the different levels of examined variables.

It should be better if the authors also change the graph's X axis to Before lockdown (BL) and After lockdown (AL). And add a title for the Y axis, line "Number of injuries"  

p.10, Table 5 title: it is suggested that the legend is written: Pairwise comparisons for the different levels of the variables moment of injury and mechanism of injury between before (BL) and after lockdown (AL) period.

Also, it is suggested that the table's Pre and Post be changed to BL and AL. 

p.10, Table 6 title: same comment as the previous one: the title needs to be more specific like "Results in the IR for  the examined variables BL and AL". also, change pre and post in the Table to BL and AL

p.12, line 401: correct to "...muscle injuries by 14.5% and tendinopathies by 8.1%..." 

p.12, lines 409-410: it is suggested to write "...(e.g., as for the Spanish national soccer leagua "La Liga")...

p.14, lines 528-533: I am afraid that I do not agree with the authors arguments about the theoretical insight that their work could provide. I accept that a lockdown could be considered as a prolonged irregular training period -not process-, however the sample of the study was rather small (n=94 respondents), the age range was limited (young adults athletes) and thus, one should be very careful about talking about theoritical insight. The practical perspective on the other hand is legitimate and it determines the significance of the study, since Spain was especially hit by CoVid19. My suggestion is that those lines are removed from the text.

p.14, line 535: correct to "have experienced..."

Author Response

Dear reviewer,

Thank you again for your suggestions and considerations about this research, your thorough review has been very insightful and has helped us to improve our manuscript. Now, we will try to answer your comments in the best way possible.  

The lines we use are those of the tracked changes version of the manuscript.

  1. page 2., line 52: correct "training load caused to..." to "training load imposed on ..."
    1. The correction has been made (L52)
  2. page 2, line 67: it is suggested that is written "...return to sport, as a result of the detraining process related effect.
    1. The change has been made (L67)
  3. page 2, line 88: correct to "alludes"
    1. The correction has been made (L87)
  4. page 4, line 138 & 140: better state "sport professionals"
    1. The term “sport” has been included in L138/140.
  5. page 5, section 2.3. This is helpful, however it is recommended that is written in a statistical way, whenever possible. For example: Type of sport: variable with 2 levels, specifically contact (....) vs non-contact sports (.....)
    1. Section 2.3 has been rewritten in a more statistical way (L183-210)
  6. page 7, line 276: add at the end of the phrase ... and level of significance was set at a=0.05.
    1. The level of significance has been included (L273-274).
  7. general comment about results: statistically speaking, in my opinion, it would be more accurate if the variable referred to two or three levels instead of conditions. In the sense that a condition implies a situation that could be modified.
    1. The term “condition” has been changed to “level” throughout the manuscript.
  8. Figure 1, legend: it is suggested to change it to: Number of injuries before (BL) and after lockdown (AL) for the different levels of examined variables.
    1. Title of figure 1 has been modified (L289).
  9. It should be better if the authors also change the graph's X axis to Before lockdown (BL) and After lockdown (AL). And add a title for the Y axis, line "Number of injuries"
    1. Figure 1 has been modified, changing X axis and adding title of Y axis (L288).
  10. 10, Table 5 title: it is suggested that the legend is written: Pairwise comparisons for the different levels of the variables moment of injury and mechanism of injury between before (BL) and after lockdown (AL) period.
    1. The title of table 5 has been modified (L340-341)
  11. Also, it is suggested that the table's Pre and Post be changed to BL and AL.
    1. Pre and Post have been changed to BL and AL.
  12. 10, Table 6 title: same comment as the previous one: the title needs to be more specific like "Results in the IR for the examined variables BL and AL". also, change pre and post in the Table to BL and AL
    1. Title of table 6 and pre/post have been changed (L358-359)
  13. 12, line 401: correct to "...muscle injuries by 14.5% and tendinopathies by 8.1%..."
    1. The correction has been made (L398).
  14. 12, lines 409-410: it is suggested to write "...(e.g., as for the Spanish national soccer league "La Liga")...
    1. The sentence has been rewritten (L406-407).
  15. 14, lines 528-533: I am afraid that I do not agree with the authors arguments about the theoretical insight that their work could provide. I accept that a lockdown could be considered as a prolonged irregular training period -not process-, however the sample of the study was rather small (n=94 respondents), the age range was limited (young adult athletes) and thus, one should be very careful about talking about theoretical insight. The practical perspective on the other hand is legitimate and it determines the significance of the study, since Spain was especially hit by CoVid19. My suggestion is that those lines are removed from the text.
    1. Theoretical and practical implications section has been modified (L523-525).
  16. 14, line 535: correct to "have experienced..."
    1. Correction has been made (L527).

We hope we have answered properly all your considerations. Again, thank you very much for your time and effort reviewing our manuscript. Best regards.

Reviewer 2 Report

The document has been greatly improved. It is still in my opinion an oversimplified design, but the exposition of results has been clarified for the reader in this new version as has the discussion.

Author Response

Dear reviewer,

Thank you again for all your suggestions and considerations about this research, your thorough review has been very insightful and has helped us to improve our manuscript. Best regards.

Reviewer 4 Report

Thank you for the possibility to review the manuscript. I am generally pleased with the improvement of the structure and content of this paper.

Major concern:
The low number of participants (n=94) from both amateur (n=41) and high level athletes (n=53) makes the generalization of the results difficult. However, as the research of the effect of lockdown on the injury rate is unique, this study may contribute to the limited knowledge in this field.

Author Response

Dear reviewer,

Thank you again for your suggestions and considerations about this research, your thorough review has been very insightful and has helped us to improve our manuscript.

About your concern about sample size, we carried out a statistical power analysis using G-Power 3.1 software, and the result was that we needed a sample size of 92 participants. In any case, we will keep your suggestion in mind for future research and will try to collect a larger sample.

Again, thank you very much for your time and effort in reviewing our manuscript. Best regards.

Reviewer 5 Report

Dear Aurhors

Great Work.

Thaks for the changes. 

Author Response

(The authors gave the same response as above.)
